# Gastric Serotonin Biosynthesis and Its Functional Role in L-Arginine-Induced Gastric Proton Secretion

**DOI:** 10.3390/ijms22115881

**Published:** 2021-05-30

**Authors:** Ann-Katrin Holik, Kerstin Schweiger, Verena Stoeger, Barbara Lieder, Angelika Reiner, Muhammet Zopun, Julia K. Hoi, Nicole Kretschy, Mark M. Somoza, Stephan Kriwanek, Marc Pignitter, Veronika Somoza

**Affiliations:** 1Department of Physiological Chemistry, Faculty of Chemistry, University of Vienna, Althanstraße 14, 1090 Vienna, Austria; ann-katrin.holik@univie.ac.at (A.-K.H.); kerstin.schweiger@univie.ac.at (K.S.); barbara.lieder@univie.ac.at (B.L.); muhammet.zopun@univie.ac.at (M.Z.); marc.pignitter@univie.ac.at (M.P.); 2Christian Doppler Laboratory for Bioactive Aroma Compounds, Faculty of Chemistry, University of Vienna, Althanstraße 14, 1090 Vienna, Austria; verena.stoeger@univie.ac.at (V.S.); julia.katharina.hoi@univie.ac.at (J.K.H.); 3Pathologisch-Bakteriologisches Institut, Sozialmedizinisches Zentrum Ost- Donauspital, Langobardenstraße 122, 1220 Vienna, Austria; angelika.reiner@wienkav.at; 4Department of Inorganic Chemistry, Faculty of Chemistry, University of Vienna, Althanstraße 14, 1090 Vienna, Austria; Nicole.kretschy@univie.ac.at (N.K.); mark.somoza@univie.ac.at (M.M.S.); 5Food Chemistry and Molecular Sensory Science, Technical University of Munich, Lise-Meitner-Straße 34, 85354 Freising, Germany; 6Leibniz Institute for Food Systems Biology, Technical University of Munich, Lise-Meitner-Str. 34, 85345 Freising, Germany; 7Chirurgische Abteilung, Sozialmedizinisches Zentrum Ost- Donauspital, Langobardenstraße 122, 1220 Vienna, Austria; stefan.kriwanek@wienkav.at; 8Nutritional Systems Biology, School of Life Sciences, Technical University of Munich, Lise-Meitner-Str. 34, 85345 Freising, Germany

**Keywords:** human gastric tumor cells, gastric serotonin release, immunofluorescence, proton secretion, energy metabolism

## Abstract

Among mammals, serotonin is predominantly found in the gastrointestinal tract, where it has been shown to participate in pathway-regulating satiation. For the stomach, vascular serotonin release induced by gastric distension is thought to chiefly contribute to satiation after food intake. However, little information is available on the capability of gastric cells to synthesize, release and respond to serotonin by functional changes of mechanisms regulating gastric acid secretion. We investigated whether human gastric cells are capable of serotonin synthesis and release. First, HGT-1 cells, derived from a human adenocarcinoma of the stomach, and human stomach specimens were immunostained positive for serotonin. In HGT-1 cells, incubation with the tryptophan hydroxylase inhibitor p-chlorophenylalanine reduced the mean serotonin-induced fluorescence signal intensity by 27%. Serotonin release of 147 ± 18%, compared to control HGT-1 cells (set to 100%) was demonstrated after treatment with 30 mM of the satiating amino acid L-Arg. Granisetron, a 5-HT3 receptor antagonist, reduced this L-Arg-induced serotonin release, as well as L-Arg-induced proton secretion. Similarly to the in vitro experiment, human antrum samples released serotonin upon incubation with 10 mM L-Arg. Overall, our data suggest that human parietal cells in culture, as well as from the gastric antrum, synthesize serotonin and release it after treatment with L-Arg via an HTR3-related mechanism. Moreover, we suggest not only gastric distension but also gastric acid secretion to result in peripheral serotonin release.

## 1. Introduction

Within the central nervous system (CNS), the monoamine serotonin (5-HT), synthesized from the essential amino acid L-tryptophan, is an important neurotransmitter associated with a manifold of different disorders, such as schizophrenia [1] and depression [2,3,4], but is also known to regulate pathways of satiation [5,6,7]. Although central 5-HT is believed to play a pivotal role in satiation, peripherally administered 5-HT had already been demonstrated to reduce food intake back in the early 1980s [8]. Moreover, our own recent human intervention trials demonstrated peripheral serotonin to correlate with energy intake and gastric emptying [9,10,11,12]. 

The hypothesis of peripheral 5-HT regulating mechanisms of satiation is supported by large quantities being synthesized in enterochromaffin cells of the gastrointestinal (GI) tract [13], which is chiefly involved in mechanisms regulating food intake. In the duodenum, activation of 5-HT receptors, e.g., the 5-HT3, regulates not only emetic pathways but also gastric motility, intestinal secretion and visceral sensitivity [14,15,16]. 5-HT is thought to mostly act locally, stimulating and sensitizing abdominal vagal nerve terminals which project to the brainstem; although under some circumstances, plasma concentrations of 5-HT may also be increased sufficiently to pass through the blood–brain barrier [17]. Peripheral serotonin released into the circulation upon gastric distension has been shown to stimulate c-fos expression in specific brain nuclei via 5-HT3 receptors in conscious rats, indicating a satiating potential of peripheral serotonin, as well [18].

Choi et al. [19] carried out investigations into the distribution of 5-HT in different cell lineages of three healthy human donor stomachs. Their analysis revealed distinct patterns of endocrine cells, showing 5-HT-positive cells in the proximal stomach and in the antrum. In addition, the 5-HT transporter (SERT) was identified in the antrum. 

From a functional point of view, gastric 5-HT might not only contribute to satiation by its response to food-induced gastric distension [18]. In this study, we hypothesize a satiating role for gastric 5-HT on gastric acid secretion that is different from the secretory effects of 5-HT released by enteroendocrine cells in the duodenum. Whereas in the duodenum, a hydrochloric acid-induced release of 5-HT has been shown to inhibit gastric acid secretion [20,21], an increase of gastric acid secretion as a response to food intake might result in gastric 5-HT release which, in turn, contributes to satiation [22]. This hypothesis is supported by findings showing that gastric acid secretion was associated with delayed gastric emptying of a protein-rich meal, and reduced subsequent food intake in cats that received intravenous bombesin [23], a peptide hormone that has been shown to reduce food intake in lean subjects [24].

From a mechanistic point of view, several cell line models have been studied to identify the cellular pathways by which 5-HT is synthesized, degraded and released in response to food ingredients by various cell types, such as neuroblastoma cells (SH-SY5Y) [25], enterochromaffin cells (QGP-1) [26] and the human colon carcinoma cell line Caco-2 [27,28,29]. Since we hypothesize a functional role of gastric 5-HT in gastric acid secretion, we first aimed at investigating the capacity of a stomach-derived cell line model, the human gastric tumor cells (HGT-1), to synthesize and release serotonin after nutrient stimulation, verify L-arginine (L-Arg)-induced serotonin secretion in human antrum specimens, and to investigate serotonin’s functional role in proton secretion as a pivotal mechanism of gastric acid secretion as a satiety signal. 

The human gastric cell line HGT-1 was established from a poorly differentiated adenocarcinoma on the posterior wall of the body of a patient’s stomach by Laboisse and co-workers in 1982 [30]. HGT-1 cells have been shown to express histamine H_2_ receptors, which have been demonstrated to mediate adenylyl cyclase activation and cAMP production [30]. Furthermore, this cell line has been shown to express acetylcholine receptors, the H^+^/K^+^ ATPase proton pump, an omeprazol-binding site possessing a K^+^ channel [31], angiotensin 1-converting enzyme [32], and transporters required for gastric acid secretion [33]. HGT-1 cells have consequently been established as a valuable cell model for assessing the effects of different food constituents, found, e.g., in coffee beverages, on proton secretion by measuring the intracellular pH value [34,35]. Here, we demonstrate this cell model’s capacity to synthesize serotonin and its subsequent secretion in response to exogenous stimuli, by incubating with L-Arg, an amino acid which has recently been reported to reduce food intake [11,36], delay and inhibit gastric emptying, and enhance gastric adaptive relaxation [37]. As 5-HT3 receptors have been suggested to be involved in gastric acid release [22], and initial experiments of this work demonstrated that HGT-1 cells are capable of proton secretion, we carried out co-incubation experiments with the 5-HT3 inhibitor granisetron [38]. Finally, serotonin release induced by L-Arg was confirmed in human antrum samples. With these results, we demonstrate that gastric serotonin not only regulates gastric emptying but also mechanisms of gastric acid secretion as key determinants of satiation targeted by dietary constituents.

## 2. Results

### 2.1. Cell Viability

The viability of HGT-1 cells was assessed using the MTT assay. None of the treatments led to a decrease in the metabolic activity (80%, Appendix A) of HGT-1 cells compared to the corresponding control (KRPH).

### 2.2. Serotonin Release

The serotonin-sensitive ELISA detected serotonin in the supernatants collected from cells treated with Krebs-Ringer-HEPES buffer (+0.1% ascorbic acid). As presented in Appendix A, serotonin release was the highest in HGT-1 cell line (1.25 ± 0.5 pg/cell) in comparison to Caco-2 cells (0.02 ± 0.01 pg/cell) and QGP-1 cells (0.03 ± 0.01 pg/cell). Serotonin concentrations were corrected for cell number and incubation volume. Cell number was evaluated using a Neubauers’ cell counting chamber. 

### 2.3. Serotonin Biosynthesis

Protein expression of aromatic amino acid decarboxylase (AADC) was tested by the capability of cellular lysate to react to 5-HTP provided in excess to serotonin. This experiment showed a rise in the determined serotonin with increasing 5-HTP concentrations (Figure 1). From this graph, a Lineweaver–Burk plot was derived, which led to a value of 0.28 mM for Km and 85 nmol mg protein-1 h-1 for v_max_.

### 2.4. Serotonin Staining 

Figure 2 shows a typical result of HGT-1 cells stained using an anti-serotonin antibody, whereas Appendix A displays an additional HGT-1 fluorescence staining experiment using another antibody.

In a second set of experiments, HGT-1 cells were seeded into either DMEM or DMEM containing 100 µM PCPA. In both cases, positive serotonin staining was observed (Appendix A). However, cells that received PCPA at seeding exhibited less pronounced serotonin staining (Appendix A). After image analysis with Image J, a fold-change of 0.73 was calculated, indicating a reduction in serotonin staining by 27% in the *p*-chlorophenylalanine (PCPA) treated group. In the absence of a blocking peptide, the primary anti-5-HT antibody was pre-incubated with pure serotonin for 30 min prior to the incubation. In this experiment, control cells receiving the pre-blocked version of the antibody showed a mean reduction in serotonin staining by 23%, suggesting that part of the antibody was prevented from binding to cellular serotonin by the pre-incubation. Figure 3 shows characteristic staining for serotonin in human benign and carcinoma stomach samples with positivity in scattered single cells.

### 2.5. Effects of L-Arg on Gene Expression, Serotonin Release, Proton Release and Gastric Motility

#### 2.5.1. Gene Expression

First, the effects of 50 mM L-Arg on the gene expression of HGT-1 cells were assessed in a genome-wide screening using customized cDNA microarrays. The concentration of 50 mM L-Arg was based on previous results showing an induction of protein secretion in HGT-1 cells [39]. The scatter plot shown in Appendix A shows a broad distribution of regulated probes, suggesting a strong impact of L-Arg treatment on gene expression. Pathway analysis using the Database for Annotation, Visualization and Integrated Discovery (DAVID) [40] suggested the expression of serotonin receptors being altered (Table 1A,B). As a result, we first assessed the expression of *HTR1A*, *2A*, *1B*, *3A*, *3B*, *3C*, *3D*, *3D*, *4* and *7*, in addition to the genes encoding the enzymes required in serotonin synthesis and the serotonin reuptake transporter SERT, encoded by *SLC6A4*. This showed *SLC6A4*, *TPH1*, *TPH2*, *HTR3C*, *HTR3D* and *HTR7* to be consistently expressed over 4–5 biological replicates. However, *HTR1B*, *HTR3B* and *HTR3E* were only detectable in some of the samples. *HTR1A*, *HTR2A*, *HTR3A* and *HTR4* were detected in none of the samples tested. Treatment of HGT-1 cells with L-Arg for 3 h regulated the gene expression of *SLC6A4*, *TPH2*, *HTR3C* and *HTR7* (Table 2).

#### 2.5.2. Serotonin Release

Appendix A shows a chromatogram of the supernatant collected from HGT-1 cells. Both L-Trp and 5-HT were detected in the supernatant, whereas the 5-HTP concentration was too low to be detected by LC-MS/MS. However, incubation of HGT-1 cells with 30 mM L-Arg increased the serotonin concentration in the cellular supernatant to 147 ± 18%, compared to cells treated with KRHB only (+0.1% ascorbic acid; Appendix A). This result was obtained by means of an ELISA, and verified by LC-MS/MS analysis, where an increase to 162 ± 48% was detected (Appendix A). No significant difference between the neutral pH control and the control adjusted to pH 9.5 was found. Similarly, no difference between L-Arg simply added to KRHB and the pH-adjusted L-Arg solution was detected (data not shown).

In the experiment using 10 µM granisetron, the L-Arg-induced serotonin release was prevented by the 5-HT3 antagonist, reducing the increase from 124 ± 23%, observed after L-Arg treatment, to 95 ± 16%, compared to the control set to 100% (*p* < 0.05). Granisetron alone did not affect the serotonin release compared to control cells treated with KRHB only (Figure 4).

#### 2.5.3. Serotonin Release in Human Antrum Samples

Figure 5 shows the serotonin release from human gastric antrum samples derived from gastric sleeve surgery. When incubated with L-Arg, a concentration of 10 mM, although neither 30 nor 50 mM, L-Arg stimulated the release of serotonin from gastric tissue by 44 ± 7% (*p* < 0.05).

#### 2.5.4. Proton Secretion

After a 10 min incubation time with 0.1 µM serotonin, HGT-1 cells showed a reduced proton secretion in comparison to the untreated control analyzed by means of intracellular proton index (IPX) (0.21 ± 0.03 vs. control −0.00 ± 0.02) (Figure 6A). Tested serotonin and L-Arg concentrations are based on physiologically relevant concentrations [41,42,43]. Incubation with 10 µM of the 5-HT3 receptor antagonist granisetron reduced the L-Arg-induced proton secretion from IPX −3.48 ± 0.13 to −3.00 ± 0.53 (*p* < 0.05, Figure 6B), while incubation with SERT inhibitor fluoxetine showed no impact (Appendix A).

## 3. Discussion

The indole amine serotonin has been described to exert a wide variety of actions in the human body in health and disease, reaching from its role in gastrointestinal function and the regulation of nausea, vomiting [44] and food intake [2,5] to several brain disorders, such as schizophrenia [1]. Several cell lines have been shown to release serotonin upon stimulation [25], among which enterochromaffin QGP-1 cells [26,29] and enterocyte-like Caco-2 cells [28,29] represent cell models of peripheral serotonin release. As several reports indicate the presence of serotonin in the stomach, we investigated the capacity of a cell line originating from an adenocarcinoma of the stomach and human antrum specimen to synthesize and consequently release serotonin upon stimulation by L-Arg, an amino acid for which satiating effects have been demonstrated [11]. Moreover, a functional role of serotonin on mechanisms regulating gastric acid secretion was elucidated.

First, we analyzed the expression of enzymes involved in the biosynthesis of serotonin. Similar to the study by Vieira-Coe et al. [27], reporting the ability of Caco-2 cells to synthesize and degrade serotonin, we determined the activity of AADC based on their method. In this experiment, we showed an increase in the serotonin concentration with increasing substrate concentrations of 5-HTP, thereby indicating the presence of functional AADC. Taken together, the presence of functional AADC suggests that HGT-1 cells are capable of synthesizing serotonin. To show that serotonin is, in fact, present in HGT-1 cells, we performed immunostaining experiments. Furthermore, human adenocarcinoma excised from the proximal stomach showed scattered single-cell staining positive for serotonin. This pattern is similar to the expression pattern of serotonin cells in healthy stomach tissue, in which only a low number of serotonin-expressing cells was found compared to other cell types present [19]. This finding is also concordant with the literature, demonstrating only a low number of scattered carcinoma cells with serotonin expression. In addition, only a very low percentage of serotonin expression in gastric adenocarcinomas has been described [45]. Regardless of the number of cells staining positive, this suggests that serotonin is present in proximal stomach samples of both healthy and diseased donors. Next, we compared the serotonin content of HGT-1 cells to cell lines, which have been described to release serotonin upon stimulation in previous studies. In the cellular supernatants of the enterochromaffin cell line QGP-1 and differentiated Caco-2 cells, serving as a cell model of enterocytes, we detected lower serotonin quantities than in the supernatant collected from HGT-1 cells. L-Tryptophan (L-Trp) has to be present in the cells in order to serve as substrate for serotonin synthesis. As the actual L-Trp level in the supernatant collected from HGT-1 cells was unknown, we analyzed L-Trp, 5-HTP and 5-HT by LC-MS/MS to exclude the Trp level in the sample exceeding the level reported to cross-react by the ELISA manufacturer. This showed L-Trp to be present, but not in concentrations high enough to interfere with the ELISA. Interestingly, we detected both L-Trp and serotonin, but not 5-HTP. This fits with the conversion of L-Trp to 5-HTP being the rate-limiting step, and, hence, 5-HTP being converted to serotonin immediately.

In addition to showing the enzymes required in the biosynthesis of serotonin on a functional level, we assessed the mRNA expression of the cultured cells, including several serotonin receptors and the serotonin reuptake receptor gene *SLC6A4* by qPCR. This demonstrated *TPH1*, *TPH2*, *SLC6A4*, *HTR3C/D* and *HTR7* to be expressed. However, we failed to quantify the gene expression of the second enzyme required in the synthesis of serotonin, AADC. As we were able to show AADC activity, and AADC is a prerequisite in the biosynthesis of serotonin, it seems likely that the AADC mRNA level was below the limit of detection of our qPCR method. In addition, mRNA expression of the receptor subtypes *HTR1A*, *HTR2A,* and *HTR3A* was below the limit of detection. However, genes encoding for serotonin receptors have been described in stomach tissue previously. A study by Van Lelyveld [46] reported very low gene expression levels of *HTR3A* and *HTR3B* in the stomach. Furthermore, the authors showed expression of *HTR3C*, *HTR3E* and *HTR4* in both antrum and fundus of healthy donor samples [46]. Among the currently known seven serotonin receptor families, 5-HT3 receptors are the only ones belonging to the family of cys-loop ligand-gated ion channel [44]. Serotonin has been suggested to modulate the secretion of gastric acid via this subtype of serotonin receptors [22]. Furthermore, a link between intragastric pH and serotonin release has been shown using perfused rat stomach [47]. In this study, at an intragastric pH of 2, the basal serotonin release into the vasculature was reported to be 10 times higher than that into the gastric lumen [47]. This may be of particular interest in HGT-1 cells, as they have previously been used to study the influence of a number of compounds on proton secretion [34,48]. Hence, we tested the effect of serotonin on proton secretion from HGT-1 cells, showing incubation with 0.1 µM serotonin to significantly reduce proton secretion, thus suggesting that serotonergic pathways may influence proton secretion in this cell line, similarly, as suggested by Lai et al. [22]. The fact that a lower concentration of 100 nM serotonin decreased proton secretion, whereas higher concentrations of 500 and 1000 nM serotonin had no effect might be explained by a hormetic response [49]. The 5-HT3 receptors are also of interest in the context of gastric emptying. An in vivo study on male Wistar rats carried out by Doihara et al. [26] reported a delay in gastric emptying by TRPA1 agonists involving a serotonergic pathway. In this study, a delay in gastric emptying was observed after administration of 1 mg × kg^−1^ serotonin. The effect of TRPA1 agonists on gastric emptying was abolished in experiments using either the TPH inhibitor PCPA or the 5-HT3 receptor inhibitor granisetron. 

Next, we tested the effect of L-Arg on serotonin release from HGT-1 cells and ex vivo samples derived from human antrum specimen. This showed, in both the ex vivo and in vitro setting, treatment with L-Arg to increase the serotonin content in the cellular supernatant collected. Although serotonin release by human antrum samples was reduced after exposure to high L-Arg concentrations of 50 mM, a lower concentration of 10 mM L-Arg increased serotonin release. Here, we hypothesize a hormetic dose-response of L-Ag on serotonin release. The increase could either be due to a rise in serotonin release or due to an impact on the serotonin transporter SERT. SERT expression has been reported in the stomach [50] and was also detected by qPCR in HGT-1 cells in our study. Our microarray experiment revealed treatment with L-Arg to have a strong impact on gene expression of HGT 1 cells. Pathway analysis with the free online tool DAVID showed the cluster with the second highest enrichment score to consist predominantly of genes involved in serotonin release. As a result, we carried out qPCR experiments, showing L-Arg to alter the gene expression of *HTR3C*, *HTR7*, *SLC6A4*, and *TPH2*. As L-Arg influenced the expression of *HTR3* and the *HTR3C* subunit has previously been described in the modulation of gastric acid secretion [22], we carried experiments into a possible link between L-Arg-induced serotonin release and the 5-HT3 receptor by performing co-incubation experiments with the 5-HT3 receptor antagonist granisetron. The increase in extra-cellular serotonin after treatment with L-Arg was prevented by co-incubation with granisetron, indicating that a 5-HT3 receptor-mediated pathway may be involved. A similar control of serotonin release by 5-HT3 autoreceptors has been shown for enterochromaffin cells from isolated intestinal segments of guinea pigs [51]. Moreover, a study showing delayed gastric emptying after serotonin administration revealed this effect to be abolished by either inhibiting the 5-HT3 receptor or TPH as well [26]. In the present study, the effect of L-Arg on proton secretion was decreased by granisetron co-incubation, pointing to an involvement of 5-HT3 receptors not only in the regulation of serotonin release, but also of the proton secretion in HGT-1 cells. 

Whether the results obtained from HGT-1 cells point to a possible regulation in vivo or not will need to be addressed in either human intervention studies or animal trials. In addition, the mechanism linking gastric acid secretion and serotonin release may be analyzed further in HTR3 knock-out models, and by identifying cellular mediators involved such as Ca^2+^ mobilization, cAMP or ERK activation. 

In conclusion, this study shows serotonin synthesis and release from HGT-1 cells as a model system for peripheral serotonin in the stomach and human antrum specimen. Furthermore, experiments using the 5-HT3 receptor antagonist granisetron suggest that this serotonin autoreceptor is involved in both proton and serotonin secretion. Overall, our data suggest a feedback mechanism by which serotonin release, known for its impact on food intake [9,10,11,12], not only reduces gastric activity upon food-stimulation, but also might contribute to satiation. To elucidate the satiating impact serotonin may have after release from endocrine cells of the stomach, animal trials and human intervention studies need to be conducted.

## 4. Materials and Methods

### 4.1. Materials

All reagents were obtained from Sigma-Aldrich (Vienna, Austria) unless stated otherwise. Anti-5HT and Anti-TPH2 antibodies were purchased from Merck-Millipore (Vienna, Austria).

### 4.2. Cell Culture

The human gastric tumor cell line (HGT-1, Dr. C. Laboisse, Nantes, France) was cultivated using Dulbecco’s modified Eagle medium (DMEM) containing 4.5 g L^−1^ glucose. The medium was supplemented with 4 mM L-glutamine, 10% fetal bovine serum and 1% penicillin/streptomycin (100 units penicillin, 171 µM streptomycin). Cells were maintained in a humidified incubator at 5% CO_2_ and sub-cultured at 90% confluence, using cells between passage 60 and 80 in experiments. Caco-2 cells were obtained from American Type Culture Collection (ATCC, Manassas, VA, USA) and maintained in DMEM containing 4.5 g L^−1^ glucose, 10% fetal bovine serum and 1% penicillin/streptomycin. QGP-1 cells were cultivated in RPMI medium supplemented with 10% fetal bovine serum and 1% penicillin/streptomycin.

### 4.3. Cell Viability Assay

HGT-1 cells were seeded in 96-well plates at a density of 1 × 105 cells per well and left to settle for 24 h. Next, the cells were incubated with Krebs-Ringer-HEPES buffer (KRHB, (10 mM HEPES, 11.7 mM Glucose, 4.7 mM KCl, 130 mM NaCl, 1.3 mM CaCl_2_, 1.2 mM Mg_2_SO_4_, 1.2 mM KH_2_PO_4_), supplemented with 0.1% ascorbic acid, pH 7.4) or serum-free DMEM for 5 min. Then the incubation solution was replaced with MTT working solution (3-(4,5-dimethylthiazolyl-2)-2,5-diphenyltetrazolium bromide, 100 µL, 1 mg mL^−1^) and incubated for 15 min at 37 °C in a humidified incubator. Finally, the MTT working solution was removed, 150 µL DMSO was added, and the absorbance was read at 570 nm. 

### 4.4. RNA Isolation and qPCR

HGT-1 cells were seeded in 24-well plates at a density of 1.5 × 10^5^ cells per well. The cells were treated with L-Arg for 3 h and subsequently washed with ice cold PBS. Further, RNA was isolated using the PeqLab total RNA kit (Peqlab, VWR, Vienna, Austria), following the manufacturer’s protocol. RNA quality and concentration were analyzed by the A260/280 absorbance ratio using the nanoquant plate for the Tecan Infinite 200 PRO Plate Reader. The purity of all samples was ensured within a ratio between 1.8 and 2.0. Then, 1 µg of the isolated RNA was reversely transcribed with the high-capacity cDNA kit (Applied Biosystems, Thermo Fisher Scientific, Vienna, Austria) The synthesized cDNA was added to a qPCR mix containing fast SYBR green master mix (Applied Biosystems, Thermo Fisher Scientific, Waltham, MA, USA) and the amplification assessed on a StepOnePlus device (Applied Biosystems, Thermo Fisher Scientific, Waltham, MA, USA). The qPCR cycles started with an initial denaturation step at 95 °C for 20 s, followed by 40 amplification cycles implying of denaturation at 95 °C for 3 s, annealing at 60 °C for 30 s, and elongation with a fluorescence measurement at 67 °C for 15 s. TATA-box binding protein (TBP), glyceraldehyde 3-phosphate dehydrogenase (GADPH) and peptidylprolyl isomerise A (PPIA) were used as endogenous controls. The sequences of all primers are provided in Appendix A. Primers were either designed using PrimerBlast or taken from PrimerBank. LinReg v2013.0. [52] was used in the data analysis. 

### 4.5. DNA Microarrays

Microarray analysis was carried out as described previously [53], using a reaction chamber allowing for the simultaneous synthesis of two microarrays [53]. HGT-1 cells were seeded in 6-well plates and left to settle for 24 h. Thereafter, cells were incubated with 50 mM L-Arg for 3 h prior to RNA isolation (RNeasy Mini Kit, Qiagen, Hilden, Germany). Microarray labelling, hybridization and analysis were carried out as described previously [25], however an Axon GenePix 4400A was used for scanning the microarrays. 

### 4.6. Aromatic Amino Acid Decarboxylase (AADC) Activity 

The method used in the assessment of the enzymatic activity of the aromatic amino acid decarboxylase was modified from Vieira-Coelho et al. [27]. Briefly, HGT-1 cells grown to confluence in a T175 flask were washed with HBSS and detached from the flask by scraping, after the addition of 1 mL HBSS. The cells were then subjected to three freeze-thaw cycles in liquid nitrogen, homogenized using a Potter-Elvehjem device, passed through a small-bore needle, and centrifuged at 4 °C (full speed, 15 min). A total of 100 µL of the supernatant was added to incubation medium (100 µL) and incubated for 15 min at room temperature. The incubation medium consisted of: tolcapone (4 µM), paragyline (300 µM), pyridoxal phosphate (240 µM), sodium borate (110 µM), Na_2_HPO_4_ (150 µM), KH_2_PO_4_ (350 µM) and HBSS. Next, 200 µL 5-HTP (0–9 mM) and ascorbic acid, giving a final concentration of 0.1%, were added. After 15 min incubation in the dark, 60 µL 6 M hydrochloric acid were added and centrifuged at 4 °C for 15 min. The supernatant was passed through a 0.2 µm filter prior to injecting 50 µL into the HPLC column. The analytical method used is described in the LC-MS/MS section of the supplementary materials. The total protein content of the cell lysate was determined by Bradford assay.

### 4.7. Serotonin Staining

Histological specimens were obtained from two patients from the Pathologisch-Bakteriologisches Institut, Klinikum Donaustadt, Vienna, Austria after signing a written informed consent that covers all diagnostic procedures on the extracted tissue. One specimen was derived from a 79-year old female patient undergoing distal partial gastrectomy for benign peptic ulcer with perforation. The other specimen was derived from a 47-year old male patient with proximal high-grade adenocarcinoma. Immunohistochemistry was performed on 5 µm-thick formalin-fixed, paraffin-embedded whole tissue sections. Slides were processed in the fully automated staining instrument Benchmark ULTRA using ultraView Universal DAB Detection Kit (Ventana Medical Systems, Oro Valley, AZ, USA). The primary antibody M0758 (clone 5HT-H209, Agilent DAKO, Santa Clara, CA, USA) was applied at 1:5 for 32 min at 37 °C. No pretreatment was performed. All counterstaining was performed with hematoxylin. Blocking experiments in order to control for unspecific staining were performed using normal mouse IgG1 serum replacing the primary antibody. HGT-1 cells were adhered on object slides and treated in a similar way. 

### 4.8. Serotonin Release

#### 4.8.1. Serotonin ELISA

Analysis of serotonin release in cell supernatants was carried out using High Sensitive Serotonin ELISA (DLD Diagnostika, Hamburg, Germany) as described previously [28]. HGT-1 cells were seeded in 24-well plates at a density of 1.5 × 10^5^ cells per well. One day after seeding, the cells were washed with pre-warmed PBS and subsequently incubated with either 200 µL KRHB (10 mM HEPES, 11.7 mM glucose, 4.7 mM KCl, 130 mM NaCl, 1.3 mM CaCl_2_, 1.2 mM Mg_2_SO_4_, 1.2 mM KH_2_PO_4_, supplemented with 0.1% ascorbic acid, pH 7.4) or 200 µL KRHB with 30 mM L-Arg for 5 min in a humidified incubator at 37 °C in the dark. To assess whether 5-HT3 receptors are involved, co-incubations with 10 µM 5-HT3 antagonist granisetron were carried out. After incubation, the supernatant was collected, diluted 1:5 with KRHB and the serotonin concentration assessed. A more detailed procedure for each cell type is given in the supplemental information.

#### 4.8.2. Serotonin Release from Human Antrum Samples

The effect of L-Arg on serotonin release in HGT-1 cell line was assessed in human gastric tissue. Explant culture protocol was conducted as described before [54]. Anonymized antrum samples were derived from gastric sleeve surgery patients after written consent and put in ice cold Krebs-Ringer-HEPES buffer immediately. Antrum slices were thoroughly washed with Krebs-Ringer-HEPES buffer, cut and weighed. Gastric tissue was placed in 6-well dishes containing 3 mL DMEM + 10% FBS 200 mM L-glutamine and 1% P/S and incubated for 1 h in an incubator at 37 °C and 5% CO_2_. Afterwards, samples were carefully washed with prewarmed PBS and transferred to a new 6-well dish containing 1 mL of 10, 30, 50 mM of L-Arg or Krebs-Ringer-HEPES buffer, supplemented with 0.1% ascorbic acid, pH 7.4 and incubated for 5 min. Finally, the supernatant was centrifuged two times for 1 min at 7000× *g* at 4 °C and stored at −80 °C until measurement. Serotonin release was assessed using a serotonin ELISA as described above.

### 4.9. Proton Secretory Activity

Proton secretory activity and corresponding data analysis were conducted by means of the pH-sensitive fluorescence dye 1,5 carboxy-seminaphto-rhodafluor acetoxymethylester (SNARF-1-AM; Life Technologies), as described previously [34,35,39]. Briefly, HGT-1 cells were seeded 24 h prior the experiment, in 96-well plates in a density of 100,000 cells/well. For incubation, cells were washed with KRPH and incubated with 3 µM SNARF dye for 30 min (37 °C, 5% CO_2_). Subsequently, cells were washed again and treated with 0.1–10 µM serotonin, 30 mM L-Arg. Furthermore, co-incubation experiments with 5-HT3 receptor inhibitor granisetron (10 µM) or SERT inhibitor fluoxetine were carried out. 1 mM histamine was tested as a positive control. The intracellular proton index (IPX) was calculated as the log2 data of the ratio between treated and untreated control (KRHP) cells. IPX values indicate the proton secretory of the cells, whereby lower IPX values demonstrate an increased proton secretion.

### 4.10. LC-MS/MS

Briefly, tryptophan, 5-hydroxytryptophan and serotonin in the cellular supernatants collected were analyzed in a single run by LC-MS/MS in positive MRM mode. The detailed method is described in the supplemental information.

### 4.11. Statistical Analysis

SigmaPlot 11 was used for statistical analysis. Data are presented as average ± SEM or standard deviation, as outlined in the corresponding figure legend. Similarly, the numbers of biological replicates and the statistic tests performed are stated in the legend of the corresponding figure or table. Treatments are shown in relation to control cells set to 1 or 100%, labelled as treated over control in the figures. Image J was used for reading out the fluorescence of single cells. Proton secretion data were subjected to Nalimov’s outlier test prior to analysis. Detailed information on the statistical tests applied is given in the individual table/figure caption. 

## 5. Conclusions

In conclusion, this study shows serotonin synthesis and release from HGT-1 cells as a model system for peripheral serotonin in the stomach and human antrum specimen. Furthermore, experiments using the 5-HT3 receptor antagonist granisetron suggest this serotonin autoreceptor to be involved in both proton and serotonin secretion. Overall, our data suggest a feedback mechanism by which serotonin release, known for its regulation of food intake, not only reduces gastric activity upon food-stimulation, but, further, might contribute to short term satiety regulation. To elucidate the satiating impact serotonin may have after release from endocrine cells of the stomach, animal trials and human intervention studies need to be conducted.

## Figures and Tables

**Figure 1 ijms-22-05881-f001:**
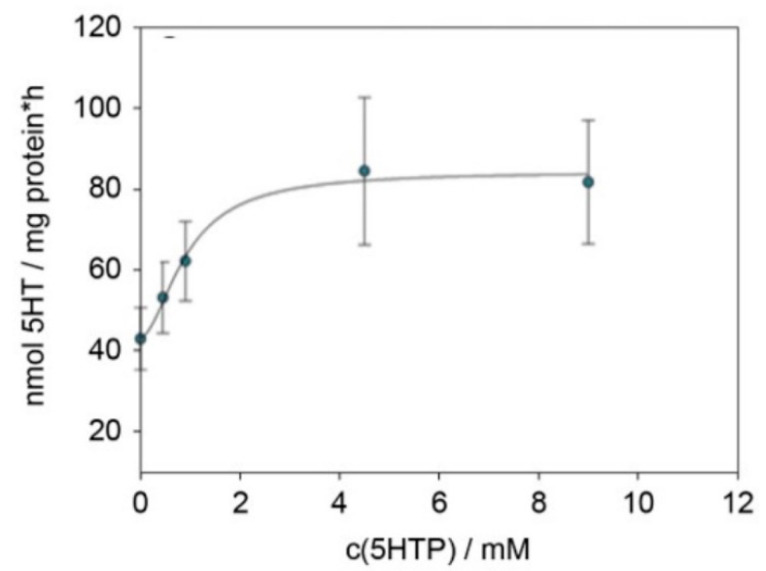
Graph of the serotonin formed in HGT-1 cells, normalized to protein concentration and incubation time vs. the substrate concentration. Data were collected from 3–4 biological replicates with 1–2 technical replicates.

**Figure 2 ijms-22-05881-f002:**
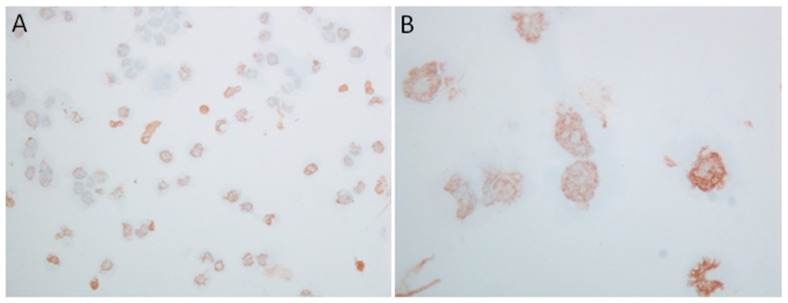
Immunohistochemical staining for serotonin in HGT-1 cells. (**A**) Original magnification ×200; (**B**) original magnification ×600.

**Figure 3 ijms-22-05881-f003:**
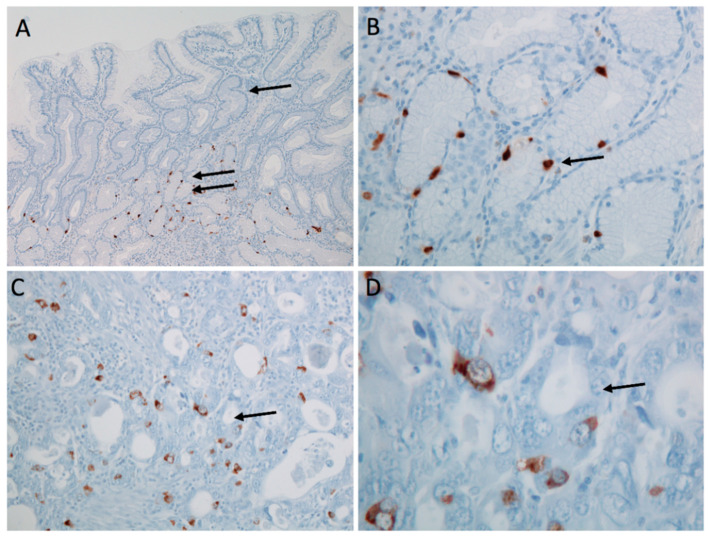
Immunohistochemical staining for serotonin in human stomach samples. Gastric antral mucosa demonstrating cytoplasmic reactivity in scattered individual enterochromaffin-like cells within gastric glands; overview (**A**); detail (**B**). Gastric adenocarcinoma with cytoplasmic reactivity in scattered individual carcinoma cells; overview (**C**); detail (**D**). Original magnification: (**A**) ×100; (**B**,**C**) ×200; (**D**) ×600.

**Figure 4 ijms-22-05881-f004:**
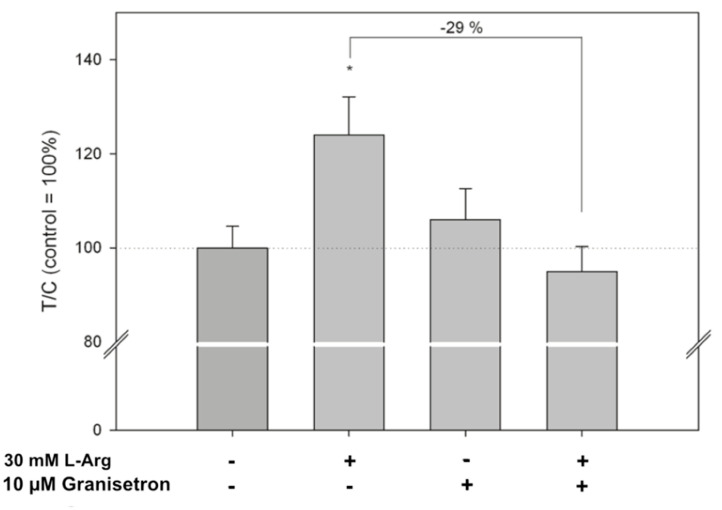
Influence of 10 µM of the 5-HT3 receptor antagonist granisetron on L-Arg-induced serotonin release by HGT-1 cells. Data are shown as average ± standard deviation of treated over control (T/C) samples, with data from L-Arg-treated cells set to 100% (T/C = 100%). *n* = 3–4, tr = 2; statistics: one-way ANOVA vs. control followed by Holm–Sidak post hoc test (*: *p* < 0.05).

**Figure 5 ijms-22-05881-f005:**
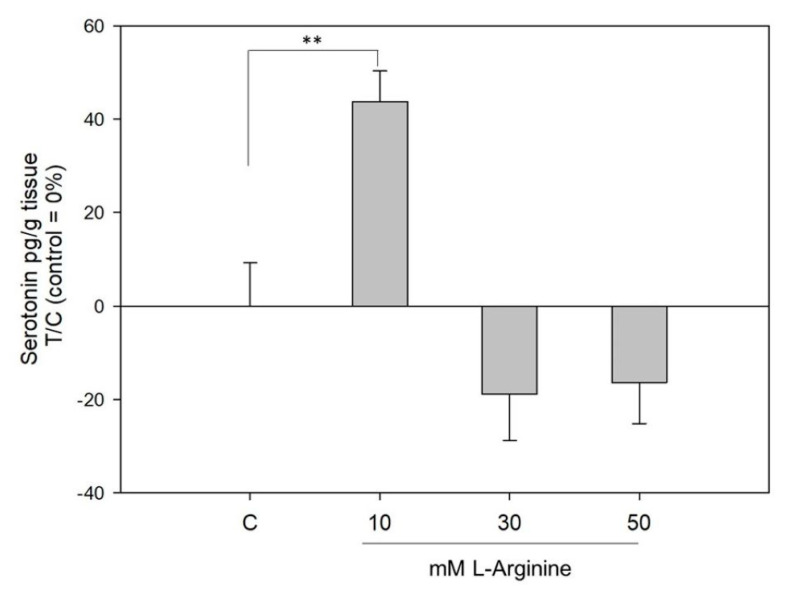
Impact of L-Arg on serotonin release in human gastric tissue. Data are presented as mean ± SEM, treated over control (=Krebs-Ringer-HEPES buffer) set to 0%. *n* = 2–3 tr = 2–4; one-way ANOVA vs. control followed by a Holm–Sidak post hoc test (**: *p* < 0.01).

**Figure 6 ijms-22-05881-f006:**
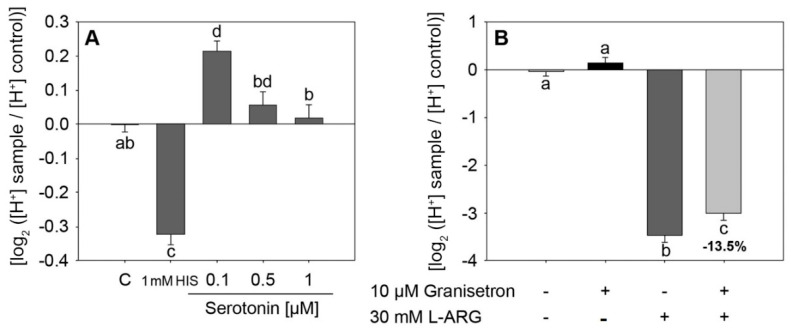
(**A**) Impact of incubation with 0.1–1 µM serotonin on proton secretion in HGT-1 cells. 1 mM HIS represents the positive control histamine in a concentration of 1 mM. *n* = 3–6, tr = 6; statistics: one-way ANOVA on ranks followed by Dunn’s post hoc test vs. control. (**B**) Impact of 10 µM 5-HT3 receptor antagonist granisetron on L-Arg-induced proton secretion, *n* = 3, tr = 3–6; statistics: one-way ANOVA followed by Holm–Sidak post hoc test. Different letters indicating statistical differences (*p* < 0.05).

**Table 1 ijms-22-05881-t001:** Clusters generated by DAVID functional annotation software using microarray data gathered from HGT-1 cells exposed to DMEM or DMEM supplemented with L-Arg for 3 h.

A: Cluster 1, Enrichment Score: 3.41		
	*p*-Value	Benjamini
disulfide bonds	1.14 × 10^−5^	3.75 × 10^−3^
glycoproteins	4.90 × 10^−4^	3.30 × 10^−2^
glycosylation site: N-linked	1.00 × 10^−3^	2.80 × 10^−1^
signal peptides	1.90 × 10^−3^	3.90 × 10^−1^
**B: Cluster 2, Enrichment Score: 3.14**		
	***p*** **-Value**	**Benjamini**
serotonin receptor signaling pathway	2.30 × 10^−7^	4.60 × 10^−4^
G-protein coupled serotonin receptor activity	2.30 × 10^−6^	1.30 × 10^−3^
serotonin binding	5.00 × 10^−5^	1.40 × 10^−2^
serotonergic synapse	1.70 × 10^−4^	1.30 × 10^−2^
5-hydroxytryptamine receptor family	8.10 × 10^−4^	4.90 × 10^−1^
neurotransmitter receptor activity	2.70 × 10^−3^	3.10 × 10^−1^
release of sequestered calcium ion into cytosol	1.50 × 10^−2^	8.90 × 10^−1^
adenylate cyclase-inhibiting G-protein coupled receptor signaling pathway	2.40 × 10^−2^	9.20 × 10^−1^
vasoconstriction	5.80 × 10^−2^	9.60 × 10^−1^
dendrite	2.10 × 10^−1^	9.30 × 10^−1^

**Table 2 ijms-22-05881-t002:** qPCR data obtained after treating HGT-1 cells with 30 mM L-Arg for 3 h. Data are shown as average ± SEM. *n* = 4–5, tr = 2–3; statistics: Mann–Whitney rank-sum test.

	Control	L-Arg	*p*-Value
TPH1	1.00 ± 0.04	1.50 ± 0.16	0.078
TPH2	1.00 ± 0.11	10.0 ± 2.0	<0.001
SLC6A4	1.00 ± 0.05	10.8 ± 1.2	<0.001
HTR7	1.00 ± 0.06	2.88 ± 0.57	0.034
HTR3D	1.00 ± 0.11	2.83 ± 0.75	0.138
HTR3C	1.00 ± 0.09	2.62 ± 0.45	<0.001

## Data Availability

The data that support the findings of this study are available from the corresponding author, upon reasonable request.

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
