# Peer review of "Gastric Serotonin Biosynthesis and Its Functional Role in L-Arginine-Induced Gastric Proton Secretion"

_ijms, 2021, doi:10.3390/ijms22115881_

Round 1
Reviewer 1 Report
This manuscript examines the ability of human gastric cells to synthesize and release (5-HT) and to respond to 5-HT with altered acid secretion. The ability to biosynthesize 5-HT has been previously demonstrated in human gastric samples and in perfused rat stomach. The effect of 5-HT on acid secretion via the 5-HT3 receptor signaling pathway has also been previously demonstrated in isolated rat stomach. However, the present study appears to be the first to demonstrate the presence of 5-HT synthesis/release in the human gastric adenocarcinoma cell line HGT-1 and to implicate L-Arg in the regulation of 5-HT release from HGT1-cells and human antrum specimens. The contribution of the present study to the research field should be more clearly specified.
While the Discussion is well and clearly written for the most part, the presentation of the results needs improvement and there are some methodological concerns that also need to be addressed. Below are some comments/recommendations.
1) There is a lack of appropriate positive and negative controls in the expression experiments.
2) The Methods section does not describe how the release of 5-HT was measured and calculated. Were the cells in these experiments cultured in serum-free medium (i.e. without exogenous 5-HT)?
3) It is necessary to include at least a brief description of the method used to measure proton secretion.
4) RT-qPCR methodology also needs to be expanded; for guidelines, see Bustin et al. 2009, Clin Chem 55, 611-22, doi: 10.1373/clinchem.2008.112797.
5) A more detailed description of the results comparing the release of 5-HT between HGT-1, Caco2 and HGT-1 cells is necessary (number of replicates, statistical tests used, statistical significance).
6) TPH1 western blots with positive control tissue should be shown.
7) Some results are not shown (“data not shown”), e. g. cell viability data, fluoxetine results. All data should be reported as main figures/tables or supplementary figures/tables.
8) How did the authors select the 5-HT and L-Arg concentrations used in the experiments?
9) The results described in lines 211-214 differ from the corresponding results in Fig. 5 – in Fig. 5, only 10 µM L-Arg increases 5-HT secretion from gastric tissue, while 30 and 50 µM L-Arg seem to have the opposite trend.
10) In Fig. 6 there is no description of statistical significance. What does "1mM HIS" mean in Fig. 6A?
11) Some descriptions of figures and tables are insufficient. Each description should include the biological sample to which it refers.
12) Supplementary figures and tables are missing from the manuscript.
Minors:
The serotonin transporter gene is referred to as SERT in some places and SLC6A4 in others, please be consistent.
Since peripheral and brain 5-HT pools are generally considered to be separated by the blood-brain barrier (BBB), please provide the original citations supporting a notion that 5-HT can cross the BBB.
Author Response
Dear Editor and reviewer, please find our rebuttal attached.

Reviewer 2 Report
Serotonin is one of the most abundant neurotransmitters in the central and peripheral nervous systems. Its role in regulating food intake is known, while other functions related to the regulation of the digestive tract are not fully understood. Therefore, taking up this topic is important and justified. I consider the selection of methods appropriate for this type of research. The introduction is correct and comprehensive. The results are discussed in an appropriate way. The materials and methods accurately describe the research technique used.
The discussion is too extensive, especially section 2 (244 - 286) should be shortened.
In addition, authors should precisely define the purpose of the research and mention the clinical aspects and the possible use of research results in a clinical context.
Figure 2, especially points C and E, are illegible and should be replaced with another one, the bar scale should be included in all photos. I also suggest adding arrows to indicate specific structures and including this in your photo description.
Similar arrows should also be placed in figure 3.

Author Response
Dear editor and reviewer, please find our rebuttal attached.

Round 2
Reviewer 1 Report
Some points have been clarified in the revised manuscript, but there are still points that need to be addressed.
First, there are some important findings that do not seem convincing and at least need to be commented on:
1) the finding that 10 mM L-Arg stimulates serotonin release (by about 40%), while slightly higher concentrations of L-Arg (30 and 50 mM) have no significant effect on serotonin release (in fact, they show an opposite trend, i.e., about 20% inhibition of serotonin release, n.s.) (Figure 5)
2) the finding that 100 nM serotonin induces a decrease in proton secretion, while higher concentrations (500 and 1000 nM) have no significant effect on proton secretion (Figure 6A).
Minor issues:
1) The authors state that the serotonin concentrations tested were based on "physiologically relevant concentrations [41-43]". The cited references report plasma serotonin levels in the range of 0.86-16.96 nM (41) or below 3 nM (42), while the authors used 100, 500, and 1000 nM serotonin in their treatments. Please clarify.
2) In the experiments comparing serotonin release between HGT -1, Caco-2 and QGP-1 cells, the concentrations of serotonin in the supernatants were "corrected for cell number and incubation volume" and expressed as pg serotonin /cell. It is a common practise that concentrations of analytes in supertanatants of adherent cells are normalised to total protein content in cell lysates. Please clarify how the cell count used to correct serotonin concentrations was determined.
3) In the experiments measuring serotonin release, HGT -1 cells were seeded in 24-well plates at a density of 150,000 cells per well, whereas in the experiments measuring proton secretion, HGT -1 cells were seeded in 96-well plates at a density of 100,000 cells per well. Is there an error in cell number or was there actually such a large difference in cellular confluence between these settings?
4) Table 1: Name the cell type
5) Figure 1: Name the cell type
6) Figure 4: Explain the meaning of "T/C" on the y-axis.; correct the sentence:” Data are shown as average ± standard deviation calculated in relation to L-Arg (set to 100%)”
7) The concluding sentence " Overall, our data suggest a feedback mechanism by which serotonin released by gastric cells upon food-stimulated proton secretion not only to reduce this gastric secretory activity but also to induce satiation." overstates the results of the study (in terms of feedback mechanism and satiety) and is also unclearly formulated.
8) The following sentences need to be corrected:
"One day after seeding, the cells were washed with pre-warmed PBS and subsequently incubated with 200 µL KRHB (10 mM HEPES, 11.7 mM Glucose, 4.7 mM KCl, 130 mM NaCl, 1.3 mM 430 CaCl2, 1.2 mM Mg2SO4, 1.2 mM KH2PO4, supplemented with 0.1% ascorbic acid, pH 7.4) or 30 mM L-Arg for 5 min in a humidified incubator at 37°C in the dark." (lines 428-32)
“Further, from IPX can be deduced to secretion activity, whereby a lower IPX indi- 461 cates to a high secretion value." (lines 461-2)
Author Response
Dear Editor,
Dear Reviewer,
Please find our detailed point-to-point response attached.
Kind regards,
Veronika Somoza
